# The Root-Colonizing Endophyte *Piriformospora indica* Supports Nitrogen-Starved *Arabidopsis thaliana* Seedlings with Nitrogen Metabolites

**DOI:** 10.3390/ijms242015372

**Published:** 2023-10-19

**Authors:** Sandra S. Scholz, Emanuel Barth, Gilles Clément, Anne Marmagne, Jutta Ludwig-Müller, Hitoshi Sakakibara, Takatoshi Kiba, Jesús Vicente-Carbajosa, Stephan Pollmann, Anne Krapp, Ralf Oelmüller

**Affiliations:** 1Department of Plant Physiology, Matthias-Schleiden-Institute, Friedrich-Schiller-University Jena, 07743 Jena, Germany; s.scholz@uni-jena.de; 2Bioinformatics Core Facility, Friedrich-Schiller-University Jena, 07743 Jena, Germany; bicj@uni-jena.de; 3Université Paris-Saclay, INRAE, AgroParisTech, Institut Jean-Pierre Bourgin (IJPB), 78000 Versailles, Franceanne.marmagne@inrae.fr (A.M.); anne.krapp@inrae.fr (A.K.); 4Institute of Botany, Technische Universität Dresden, 01217 Dresden, Germany; jutta.ludwig-mueller@tu-dresden.de; 5Graduate School of Bioagricultural Sciences, Nagoya University, Nagoya 464-8601, Japan; sakaki@agr.nagoya-u.ac.jp (H.S.); takatoshi.kiba@riken.jp (T.K.); 6Centro de Biotechnología y Genómica de Plantas, Instituto Nacional de Investigación y Tecnología Agraria y Alimentación (INIA), Universidad Politécnica de Madrid (UPM), Campus de Montegancedo, 28223 Madrid, Spain; jesus.vicente@upm.es (J.V.-C.); stephan.pollmann@upm.es (S.P.); 7Departamento de Biotecnología-Biología Vegetal, Escuela Técnica Superior de Ingeniería Agronómica, Alimentaria y de Biosistemas, Universidad Politécnica de Madrid (UPM), 28040 Madrid, Spain

**Keywords:** *Piriformospora indica*, nitrogen starvation, nitrogen metabolism, nitrate transporter, ammonium transporter, amino acid transporter, endophyte

## Abstract

The root-colonizing endophytic fungus *Piriformospora indica* promotes the root and shoot growth of its host plants. We show that the growth promotion of *Arabidopsis thaliana* leaves is abolished when the seedlings are grown on media with nitrogen (N) limitation. The fungus neither stimulated the total N content nor did it promote ^15^NO_3_^−^ uptake from agar plates to the leaves of the host under N-sufficient or N-limiting conditions. However, when the roots were co-cultivated with ^15^N-labelled *P. indica*, more labels were detected in the leaves of N-starved host plants but not in plants supplied with sufficient N. Amino acid and primary metabolite profiles, as well as the expression analyses of N metabolite transporter genes suggest that the fungus alleviates the adaptation of its host from the N limitation condition. *P. indica* alters the expression of transporter genes, which participate in the relocation of NO_3_^−^, NH_4_^+^ and N metabolites from the roots to the leaves under N limitation. We propose that *P. indica* participates in the plant’s metabolomic adaptation against N limitation by delivering reduced N metabolites to the host, thus alleviating metabolic N starvation responses and reprogramming the expression of N metabolism-related genes.

## 1. Introduction

Nitrogen is a key mineral nutrient playing a crucial role in plant growth and development [1,2,3,4,5]. The soil microbiome contributes to nitrogen acquisition, and among the best studied endosymbiotic interactions are those with N-fixing rhizobia and arbuscular mycorrhizal (AM) fungi. Legumes gain access to N through symbiotic association with rhizobia, which convert N_2_ gas into ammonia in nodules. Although several efforts have been made to incorporate biological N fixation capacity into non-legume plants [6], agricultural crop production without N fertilization is currently not conceivable. AM fungi help plants in nutrient acquisition and much progress has been made in understanding the molecular basis of P and N transfer from the fungal partner to the host plant (cf. [7]). Less is known about endophytes, although they show relatively little host specificity and have therefore a great potential for agricultural applications [8,9].

A well-studied endophytic fungus is *Piriformospora* (*Serendipita*) *indica*, which interacts with numerous host plants and promotes their growth and resistance against biotic and abiotic stresses [10,11]. The stimulation of the growth of its hosts suggests that the fungus promotes nutrient acquisition, including nitrogen. An effect of *P. indica* on nitrate uptake and the nitrogen metabolism in the hosts has been reported repeatedly. On a full medium, the fungus promotes nitrogen accumulation and the expression of nitrate reductase in *Arabidopsis thaliana* [12]. In sunflower, *P. indica* increases the absorption of nitrogen by the root [13]. Strehmel et al. [14] showed that the concentration of nitrogen-rich amino acids decreased in inoculated *A. thaliana* plants. Ghaffari et al. [15] proposed that the nitrogen metabolism plays an important role in systemic salt-tolerance in leaves of *P. indica*-colonized barley. Furthermore, Lahrmann et al. [16] showed that the *P. indica* ammonium transporter Amt1 functions as a nitrogen sensor mediating the signal that triggers the in planta activation of the saprotrophic program. In Chinese cabbage, the amino acid γ-amino butyrate in particular is de novo synthesized in colonized roots [17]. Bandyopadhyay et al. [18] demonstrated that *P. indica*, together with *Azotobacter chroococcum*, facilitates the higher acquisition of N and P in rice. *P. indica* also improves chickpea productivity and N metabolism in a tripartite combination with Mesorhizobium [19]. Finally, *Serendipita williamsii* does not affect P status but C and N dynamics in AM tomato plants [20]. These examples highlight the importance of the N metabolism on numerous beneficial effects of *P. indica* for different plant species; however, how the fungus influences the host N metabolism is not clear. In this study, we use the model plant *A. thaliana* to investigate how *P. indica* interferes with N uptake and metabolism under N-limiting conditions.

## 2. Results

### 2.1. Shoot Growth Promotion by P. indica Requires External N Supply

*P. indica* colonizes *A. thaliana* roots and induces the visible growth promotion of *A. thaliana* seedlings after 4–7 days in full medium [21]. After 5 days, the fresh weight of the shoots was significantly increased (+41.9%) by the fungus, while barely any growth promotion was detectable on N-limited medium (Figure 1A). Root growth was neither affected by the fungus nor by N availability (Figure 1B). We conclude that under these experimental conditions, the shoot growth but not the root growth of *A. thaliana* seedlings is promoted by *P. indica*, and this requires N in the medium.

### 2.2. P. indica Colonisation Did Not Change the Total N Content in the Shoots and Transfer of ^15^N from the Medium to the Shoots

To test whether *P. indica* interferes with N accumulation or uptake into the plant under N-limiting conditions, the total N content in the shoots and the amount of ^15^N from ^15^NO_3_^−^-labelled growth medium in the shoots were compared for uncolonized and colonized seedlings, grown on either full or N-limiting media. As expected, the total N content in the shoots of seedlings that were exposed to N limitation was lower than in the shoots of seedlings grown on full medium (Figure 2A). Furthermore, the accumulation of ^15^N in the shoots was much higher on full medium than on medium with low N (Figure 2B). However, we did not observe significant differences for uncolonized and colonized seedlings. This suggests that the fungus does not stimulate nitrate uptake from the medium under N-sufficient and N-limiting growth conditions.

N limitation might influence the colonisation of the roots. We observed that roots on N-limiting conditions were around two times more colonized than roots on full medium (Appendix A), although the difference was not significant. This indicates that in spite of a higher colonisation rate, the transport of ^15^N label from the ^15^NO_3_^−^-containing medium to the leaves was not stimulated by the fungus under N-limiting conditions.

### 2.3. ^15^N Label Is Transferred from P. idica to the Host under N-Limiting Conditions

Since *P. indica* did not promote NO_3_^−^ uptake, we tested whether labelled ^15^N metabolites are translocated from the fungus to the plant. As shown in Figure 3A, *P. indica* was cultured on ^15^N-containing medium for 14 days before co-culturing with *A. thaliana* seedlings on full or N-limited media.

The ^15^N-labelled fungal mycelium was positioned about 1 cm away from the roots. Establishing contact between the two partners and the initiation of root colonisation started approximately 24 h later, after the growing hyphae have reached the roots (Figure 3A). Since the label could be detected in the aerial parts of all analysed seedlings, which were in contact with *P. indica*, the fungus transfers N-containing metabolites to the roots of its host, and the label is further translocated to the aerial parts of the seedlings (Figure 3B). Interestingly, more ^15^N accumulated in the aerial parts of the plants under N-limiting conditions although the data were not significantly different (Figure 3B). This suggests that the fungus helps the host with reduced N metabolites to compensate N limitation during growth on NO_3_^−^-limiting medium.

### 2.4. Reprogramming of the Metabolite Profiles to N Limitation Conditions Is Alleviated by P. indica

Next, we tested whether the fungus affects a host’s N metabolism under sufficient N and N-limitation conditions. We measured the levels of primary metabolites using GC-MS in the rosettes after 2 days of transfer to N-limitation conditions compared to N-sufficient conditions in the absence and presence of *P. indica* (Appendix A). We then calculated the metabolite ratios for plants grown under limiting versus sufficient N and compared these ratios for plants grown in the absence and the presence of the fungus. Although the amino acid profiles were comparable in colonized and uncolonized shoots, we observed slight differences for several amino acids (Table 1A). The content of aspartate and alanine decreased under N-limiting conditions in both the absence and the presence of the fungus, and these decreases were less pronounced in colonized shoots (Appendix A). Similar tendencies were observed for amino acid content that increased under N-limitation conditions; these increases were less distinct in the presence of the fungus in the case of isoleucine, lysine, tryptophan, phenylalanine, leucine, and arginine (Appendix A). The alterations in serin contents that were triggered by N limitation varied strongly in colonized plants in comparison to uncolonized plants. We then analysed the effect of N-limitation on soluble sugars (Table 1B). In the presence of *P. indica*, N-limitation triggered stronger increases in monosaccharides—in particular in glucose and fructose. The stress-related sugars trehalose and raffinose showed strong variations between the three independent replicates. However, raffinose tended to accumulate at higher levels under N limitation when the roots were colonized (see the Section 3). Overall, the slight alteration of the metabolite profiles in response to N-limitation by the colonisation with *P. indica* suggest a lessening of the effects of N limitation on several steps of the central metabolism.

### 2.5. P. indica Stimulates Expression of Specific Host´s Transporter Genes under N Limitation

The incorporation of ^15^N from the agar plate into the aerial parts of colonized seedlings is lower under N starvation when compared to seedlings grown on full medium (Figure 2), while a stimulation is observed for the translocation of labelled ^15^N from *P. indica* to the leaves under N limitation (Figure 3). To test whether genes for N metabolite transporters are regulated by *P. indica*, we performed expression profiles with RNA from the roots and shoots of seedlings, which were either grown on full or N-limited medium in the presence or absence of *P. indica* (Table 2).

Of the 56 investigated genes, which code for NO_3_^−^, NH_4_^+^, amino acid or peptide transporters, 33 genes were differentially expressed in either the roots and shoots or both of colonized and uncolonized seedlings grown on full or N-limited media (Table 2). In the shoots, this included genes for two NH_4_ transporters (*AMT1-3* and *AMT1-5*), three NO_3_^−^ transporters (*NRT2.2*, *NRT2.4* and *NRT2.5*), five members of the NITRATE TRANSPORTER 1/PEPTIDE TRANSPORTERs gene family (*NPF2.6*, *NPF2.13*, *NPF5.3*, *NPF5.12* and *NPF5.14*) as well as the urea transporter *DUR3*. Furthermore, 21 amino acid transporters, including members of the LHT and AAP families, as well as 12 UmamiT putative amino acid transporters responded to the fungus. In contrast, seven transporter genes were downregulated by *P. indica* in the shoots of N-starved seedlings (Table 2). In the roots, six of these genes showed the same regulation (Table 2). This clearly demonstrates that the expression of genes for NO_3_^−^, NH_4_^+^, amino acid and peptide transporters are major targets of the fungus under N-limited conditions (see the Section 3).

## 3. Discussion

N limitation has severe consequences for plant performance [22], and endophytes may help plants to better adapt to the shortage. We used the well-investigated symbiotic interaction between the model plant *A. thaliana* and *P. indica* to address this question. We demonstrate that under severe N limitation, the fungus does not stimulate the uptake of nitrate into the host plant but rather the N-label from fungal metabolites appears in the leaves of the host. Since our N-limiting medium contains barely any nitrate, the absence of a detectable stimulatory effect of the fungus on nitrate uptake into the host is not surprising. The N metabolites that are translocated from the hyphae to the plants under N limitation conditions did not results in fungus-induced growth promotion (Figure 1), suggesting that the N supply to the host by the fungus might only compensate deficits. Furthermore, N-translocation from the fungus to the host occurs only under N limitation conditions, suggesting the involvement of an N-sensing system (cf. [16]). The successful transfer of ^15^N by arbuscular mycorrhizal fungi to host plants has been shown previously [23,24,25]. More recently, Hoysted et al. [26] investigated clover (*Trifolium repens*) colonized by Mucoromycotina fungi and showed that the host gained both ^15^N and ^33^P tracers directly from the fungus in exchange for plant-fixed C. Whether the N supply to the host in our study system has comparable symbiotic features with profit for both partners or whether it is just the stress-related withdrawal of N from the fungus by the plant without any profit for the microbe remains to be investigated. However, since the fungus can grow and propagate on the host under our –N conditions, the N translocation to the host does not restrict hyphal growth. It appears that the conditions are not strong enough to induce changes in the symbiotic interaction [16]. It is also not clear which metabolites are transported from the microbe to the plant or how this occurs. In *Medicago truncatula*, three AMT2 family ammonium transporters (AMT2;3, AMT2;4, and AMT2;5) are involved in the uptake of N in form of ammonium from the periarbuscular space between the fungal plasma membrane and the plant-derived periarbuscular membrane [27]. In exchange, host plants transfer reduced carbon to the fungi [28,29,30]. Additionally, Cope et al. [31] showed that the colonization of *M. truncatula* with *R. irregularis* led to the elevated expression of the mycorrhiza-induced *AMT2;3* and the nitrate transporter *NPF4.12* as well as the putative ammonium transporter *NIP1;5* in the roots. A dipeptide transporter from the arbuscular mycorrhizal fungus *Rhizophagus irregularis* is upregulated in the intraradical phase [32]. To investigate how the fungus manipulates the host N metabolism, we performed a comprehensive metabolome and transcriptome analysis for N-related metabolites and genes (Table 1 and Table 2).

No major impact of the colonization by *P. indica* on the changes of shoot metabolite levels in response to N limitation has been observed in this study. Liu et al. [33] demonstrated that raffinose positively regulates maize drought tolerance by reducing leaf transpiration. The raffinose family oligosaccharides are associated with various abiotic and biotic stress responses in different plant species (e.g., [34,35,36,37,38]). It is conceivable that the stimulatory effect of *P. indica* on the raffinose level in N-limited leaves reduces stress.

Nitrate transporter genes are often upregulated under N starvation; however, the role of endophytic microorganisms in nitrate acquisition is not fully understood. In rice, the arbuscular mycorrhizal fungus *R. irregularis* remarkably promoted growth and N acquisition, and about 42% of the overall N could be delivered via the symbiotic route under nitrate-limiting conditions [39]. Nitrate uptake occurs via NITRATE TRANSPORTER1/PEPTIDE TRANSPORTER FAMILY (NPF)4.5, a member of the low affinity nitrate transporter family, which is exclusively expressed in the arbuscles of the Gramineous species [39]. A comparable mechanism does not exist in our endophyte/*A. thaliana* model, and the putative *A. thaliana* NPF4.5 homolog is not upregulated in colonized roots under N-limited conditions. However, we observed a highly specific response of several NPF/NRT1 and NRT2 family members to *P. indica* colonisation, which suggest conclusions regarding how the fungus interferes with the plant N metabolism. The nitrate transporters NRT2.2 and NRT2.4 [40] are only upregulated in the rosettes when the roots are colonized by *P. indica*, while their expression in the roots does not respond to the fungus. This suggests that the fungus promotes nitrate scavenging, which is released from the vacuole in response to N starvation. In fact, NRT2.4 has been shown to be expressed close to the phloem in rosettes and to contribute to nitrate homeostasis in the phloem under limiting nitrate supply, since in nitrate-starved *nrt2.4* mutants, nitrate content in shoot phloem exudates was decreased [40]. Likewise, *NRT1.7* (*NPF2.13*) and *NRT1.8* (*NPF5.3*) are upregulated by *P. indica* in leaves, but not in roots. NRT1.7 loads excess nitrate stored in source leaves into phloem and facilitates nitrate allocation to sink leaves. Under N starvation, the *nrt1.7* mutant exhibits growth retardation, indicating that the NRT1.7-mediated source-to-sink remobilization of stored nitrate is important for sustaining growth in plants [41]. *NRT1.8* is expressed predominantly in xylem parenchyma cells within the vasculature and functional disruption of *NRT1.8* significantly increased the nitrate concentration in xylem sap [42]. In contrast, *NRT2.3 and -2.6* are downregulated under N-limiting conditions and this is further promoted by the fungus. *NRT2.6* has been linked to biotic and abiotic stress responses [43], and it appears that the downregulation of *NRT2.6* expression by *P. indica* alleviates the stress responses in the roots. Finally, *NRT1.9 (NPF2.9)* is strongly downregulated by *P. indica* in the leaves. *NRT1.9* is expressed in the companion cells of phloem. In *NRT1*.9 mutants, downward nitrate transport was reduced, suggesting that *NRT1.9* facilitates the loading of nitrate into the phloem and enhances downward nitrate transport to the roots [44], apparently a process that is restricted by the fungus. Taken together, the analysis of the regulation of the *A. thaliana* nitrate transporter genes by *P. indica* suggests that the root-colonizing fungus supports nitrate transport to and availability in the aerial parts of the host under our nitrate-limiting conditions. This is further supported by the upregulation of *NRT1.15* (*NPF5.14*) by *P. indica* in the leaves. *NRT1.15* is a tonoplast-localized low-affinity nitrate transport [45] and the overexpression of the gene significantly decreased vacuolar nitrate contents and nitrate accumulation in *A. thaliana* shoots. *NRT1.15* regulates vacuolar nitrate efflux, and the reallocation might also contribute to osmotic stress responses other than mineral nutrition [45].

Since the medium does not contain NH_4_^+^, the plant can only receive NH_4_^+^ from the fungus via ammonium transporters (AMTs) [41,46,47,48]. *AMT1-4* expression is upregulated by *P. indica* in roots under N-limited conditions. Since AMT1-4 is root-specific [47], this suggests that the plant tries to compensate its N limitation by stimulating NH_4_^+^ uptake. NH_4_^+^ might also originate from the fungus, and it is conceivable that withdrawal of this ion from the fungus might ultimately result in a change of the symbiotic interaction towards saprophytism (cf. [16]). Furthermore, the expression of *AMT1-3* and *AMT1-5* as well as *DUR3*-coding for a urea transporter [49,50] is stimulated by *P. indica* in the shoots. Root colonization might create a metabolite environment in the host that requires these transporters for the proper distribution of the N metabolites in the aerial parts.

Seven amino acid transporters are regulated >log2-fold by *P. indica* colonisation in nitrate-deprived *A. thaliana* seedlings. In roots, the fungus prevents the downregulation of the gene for glutamine secreting GLUTAMINE DUMPER (GDU)1 [51], suggesting that the microbe wants to access to the plant glutamine. Furthermore, the broad-specificity high affinity amino acid transporter LYSINE HISTIDINE TRANSPORTER (LHT)1 [52], AMINO ACID PERMEASE (AAP)4, γ-AMINOBUTYRIC ACID TRANSPORTER (GAT)1, and CATIONIC AMINO ACID TRANSPORTER (CAT)5 are upregulated in the leaves of *P. indica*-colonized seedlings. These transporters have been proposed to be involved in nitrogen recycling in plants [53]. Apparently, a better or different N metabolism management is required for the plant when the roots are associated with the endophyte. LHT1 and -2 are also involved in the transport of 1-aminocyclopropane carboxylic acid, a biosynthetic precursor of ethylene [54], which might indicate an increased stress caused by the interaction with the fungus under N-limiting conditions. An involvement in nitrogen recycling has also been proposed for 5 of the 10 USUALLY MULTIPLE ACIDS MOVE IN AND OUT TRANSPORTERS (UMANIT13, −20, −40, −45 and −47) [53], which are regulated >log2-fold in either the roots or shoots of *P. indica*-colonized seedlings under N-limitations.

## 4. Materials and Methods

### 4.1. Plant and Fungus Material and Corresponding Growth Conditions

*A. thaliana* seeds (*Col-0*) were surface-sterilized and sown on N-free MGRL medium supplemented with 2.5 mM NH_4_NO_3_ and 3 g/L gelrite [55]. The KNO_3_ and Ca(NO_3_)_2_ in the MGRL medium were replaced by KCl and CaCl_2_ to ensure ion equilibrium. After 48 h of stratification at 4 °C in the dark, the seeds were transferred to long-day conditions with 22 °C, 16 h light/8 h dark, 80 μmol m^−2^ s^−1^ for 10 days.

*Piriformospora indica* was cultured on Kaefer’s medium as described previously [56,57]. As described previously, plugs of a 4-week-old fungal culture were used for co-cultures with the seedlings. The fungus was pre-grown for 7 days on PNM medium (PNM+N) with a nylon membrane in the dark at 22 °C. For N-limiting conditions (0 mM total N, PNM−N), KNO_3_ and Ca(NO_3_)_2_ were replaced by KCl and CaCl_2_. For control plates without fungus, only empty KM plugs were placed on top of the membrane.

### 4.2. Plant-Fungus Co-Cultures and the Determination of Growth Promotion

For plant-fungus co-cultures for 5 days, 4 plants (per Petri dish) were placed on top of the pre-grown fungal lawn, as described previously [57], with some adaptations. Plates were sealed with 3M^TM^ Micropore tape to reduce the condensation and 10-day-old plants were used for co-cultivation to reduce the amount of N, which accumulated in the plants on MGRL medium before the co-culture. In pilot experiments, we showed that the reduced age did not affect the establishment of the symbiosis with the fungus. The co-cultures were incubated at 22 °C, 16 h light/8 h dark and 80 μmol m^−2^ s^−1^ with light from the top.

After 5 days, the roots and shoots of the plants were harvested separately. For that, 5 plates (=20 plants) were harvested as 1 sample. Both roots and shoots were washed in sterile distilled water and carefully dried before weighing and direct freezing in liquid nitrogen. Samples were stored in −80 °C until further use. These experiments were repeated 3–4 times independently.

To determine growth promotion by the fungus, the weight of the sample with fungus was normalized (divided) to the weight without fungus. This was performed for the total weights sampled from the full medium (PNM+N) as well as from the N-limited medium (PNM−N). Final growth promotion values presented in the figures are averages of 3 replicates from independent cultures.

### 4.3. ^15^N Labelling Experiments in the Medium

To analyse the uptake of nitrogen by the plant, 2.5% of the total KNO_3_ (which equals 0.125 mM KNO_3_) of the PNM medium was replaced by K^15^NO_3_ (Eurisotop, Saint-Aubin, France) dissolved in distilled water. For proper comparison, the 2.5% of K^15^NO_3_ was also added to the N-free medium (PNM−N) resulting in a final concentration of 0.125 mM nitrate. Finally, PNM−N control plates without ^15^N were used and contained 0.125 mM unlabelled KNO_3_. Plants grown on these plates were compared to those grown on PNM+N plates to analyse the natural abundance of ^15^N in the plant tissue. As described before, the fungus or control plug was pre-incubated on the PNM with nylon membrane for 1 week before plants were placed on the plates. The co-cultures were incubated for 5 days to ensure that enough ^15^N was taken up by the plant.

### 4.4. ^15^N Fungus-Labelling Experiments

To analyse whether the fungus can directly transfer N or N-containing metabolites to the plant, it was labelled with ^15^N before the co-culture. A modified KM medium without the N-containing components (20 g/L dextrose, 50 mL/L macronutrients, 10 mL/L micronutrients and 1 mL/L Fe-EDTA, 1 mL/L vitamin mix, pH 6.5) was prepared and supplemented with 10 g/L ISOGRO^®^-15N (CortecNet, Les Ulis, France) according to the manufacturers protocol. *P. indica* plugs of 2 mm diameter were incubated (23 °C, 50 rpm, dark) in 2 mL of KM^ISOGRO^ in Greiner CELLSTAR^®^ 12-well plates (Greiner Bio-One, Frickenhausen, Germany) sealed with 3M^TM^ Micropore tape. After 14 days of growth, the fungal tissue was separated from the remaining medium and carefully washed 3 times with N-free liquid PNM to remove the ^15^N bound to the hyphal surface. A 76.66% enrichment in ^15^N was achieved using this protocol. The fungus was carefully cut in 5 × 5 mm pieces and placed on PNM−N and PNM+N plates to start the co-cultures. To minimize ^15^N uptake by the plant from dead fungal material during the washing and handling procedure, the fungal plugs were placed in minimum 1 cm distance from the roots. Under these conditions, contact between the two symbionts requires the active growth of the hyphae towards the roots. Co-cultivation was performed with 3 plants per plate for 14 days to ensure that enough ^15^N was taken up by the plant.

### 4.5. Isolation and Clean-Up of RNA

RNA was isolated from the roots or shoots of 10-day-old seedlings, which were either co-cultured with the fungus for additional 5 days (root colonization results), or for an additional 4 days for expression profiling. Samples of root or shoot material were stored in −80 °C. For homogenization, the samples were ground with mortar and pistil in liquid nitrogen. A maximum of 100 mg material was used for RNA extraction. RNA was extracted with Trizol^TM^ (ThermoFisher Scientific, Waltham, MA, USA) and chloroform according to the manufacturers protocol. Briefly, the plant material was mixed with 1 mL of Trizol^TM^ and incubated on a shaker at room temperature for 15 min. After the addition of 250 μL chloroform and a second incubation phase, the sample was centrifuged (30 min, 4 °C). The supernatant was mixed with isopropanol and incubated on ice, followed by centrifugation. The pellet was washed twice with 80% ethanol, dried and resuspended in RNAse-free water. The RNA isolation was followed by an additional cleaning step to remove access salts originating from the fungus tissue. For this, the sample was mixed with 3 M sodium acetate (1/10 (*v*/*v*) in RNAse-free water, pH = 5.2) and 600 µL of ice-cold 100% ethanol and incubated at −20 °C for at least 1 hr. After centrifugation and 2 cleaning steps with 80% ethanol, the sample was resuspended in RNAse-free water. The quality and concentration of the extracted RNA was tested via absorbance analysis using a NanoVue (GE Healthcare, Uppsala, Sweden).

### 4.6. RNAseq and Data Analysis

After the transfer of samples to Novogene Genomics Service (Cambridge, UK), the RNA sample integrity was checked with a Bioanalyzer 2100 (Agilent). After samples passed the quality check, the service laboratory proceeded with the library construction and RNA sequencing (PE150) on Illumina NovaSeq™ 6000 platforms, as described in a previous study [58].

The RNAseq libraries were filtered and quality-trimmed with fastp (v0.23.2) [59], i.e., read ends were truncated to achieve a Phred quality score of 30 or more. Reads below 15 nt length or those comprising at least 2 ambiguous N bases were removed from the dataset. Read qualities were monitored by FastQC (v0.11.3; https://www.bioinformatics.babraham.ac.uk/projects/fastqc/, accessed on 15 December 2022). Hisat2 (v2.2.1) [60] was used with default parameters to map the quality-trimmed RNAseq libraries to the *A. thaliana* reference genome (TAIR10, Ensembl release 51). The mapping allowed spliced reads and single-read mapping to multiple best-fitting locations. FeatureCounts (v1.5.3) [61] was applied to perform read-counting based on the *A. thaliana* reference annotation (TAIR10, Ensembl release 51). The Bioconductor DESeq2 (v1.10.0) package [62] was utilized to identify DEGs in the different pairwise mutant and wild type comparisons. Benjamini and Hochberg’s false discovery rate (FDR) approach [63] was employed to adjust the calculated *p*-values for multiple testing.

To identify DEGs of transporters predicted to transport major N compounds, the obtained results were initially filtered according to their *p*-value (*p* < 0.05). Next, DEGs were sorted according to their log2-fold change—here only changes with numbers ≥ +1.5 and ≤ −1.5 were further analysed. This list was cross-checked with targets identified from a search in the UniProt database (https://www.uniprot.org, accessed on 15 December 2022) using keywords like “NH_4_ transport”.

### 4.7. Analysis of Fungal Colonization via qPCR

A total of 1 mg of RNA was used for the synthesis of cDNA. The Omniscript RT Kit (Qiagen, Hilden, Germany) was used according to the manufacturers protocol with the oligo(dT)_18_ primer (ThermoFisher Scientific, Waltham, USA). qPCR was performed with fifty nanograms of the synthesized cDNA as template in a Bio-Rad CFX96 Real-Time PCR Detection System (Feldkirchen, Germany) via DreamTaq Polymerase (ThermoFisher Scientific, Waltham, USA) and Evagreen (Biotium, Fremont, CA, USA). The data were normalized with respect to the *A. thaliana RPS18B* (*At*1g34030) gene using the 2^−∆∆CT^ method [64]. To quantify the *P. indica* colonization level of *A. thaliana* roots, the expression of *PiTEF1* [65] was analysed in comparison to the plant´s housekeeping gene *RPS18B* (At1g 34030). The following primers were used: *PiTEF1*: CGCAGAATACAAGGAGGCC and CGTATCGTAGCTCGCCTGC; *RPS18B*: GTCTCCAATGCCCTTGACAT and TCTTTCCTCTGCGACCAGTT [66]. The colonization was compared between plants grown on PNM−N and PNM+N media (set as 1.0) using the 2^−∆∆CT^ method.

### 4.8. Determination of Total Nitrogen and ^15^N Enrichment

Total N and ^15^N contents were quantified on 1–2 mg aliquots of dry tissue, after drying a ground tissue aliquot at 65–70 °C for at least 48 h. N elements were detected using gas chromatography on a FLASH 2000 Organic Elemental Analyzer (Thermo Fisher Scientific, Villebon, France). The ^15^N/^14^N isotopic ratio was subsequently quantified using a coupled mass spectroscope (Delta V advantage IRMS; Thermo Fisher Scientific, Villebon, France). The total N content was only determined in plant shoots because the discrimination of N from plant or fungus was not possible in colonized root material.

### 4.9. Metabolomic Analysis

For GC-MS-based quantifications, 25 mg of finely ground plant material was resuspended in 1 mL of frozen (−20 °C) water:acetonitrile:isopropanol (2:3:3, *v*/*v*/*v*) containing Ribitol at 4 ug/mL and analysed as described in [67].

## 5. Conclusions

We observed an unexpected complexity in the plant N metabolism when N-deprived *A. thaliana* seedlings were colonized by *P. indica*. Our data suggest that the fungus neither stimulated the total N content nor promoted ^15^NO_3_^−^ uptake from agar plates to the host. Rather, reduced N metabolites were transported from the fungus to the plant. Furthermore, gene expression and metabolite profiles suggest that N-containing metabolites were redistributed by *P. indica* in *A. thaliana* seedlings exposed to N-limitation.

Our initial observations highlight a few aspects that need to be investigated in greater detail. (1) Which N metabolites are transported from the fungus to a plant suffering under N limitation? (2) The plant appears to adapt its N metabolism under N limitation by transporting N metabolites shootwards, a process that is supported by the fungus. Is the fungal support for the plant specific for the symbiotic phase of the interaction? (3) Are our observations *P. indica*-specific or do they occur also in other endophyte/plant interactions?

## Figures and Tables

**Figure 1 ijms-24-15372-f001:**
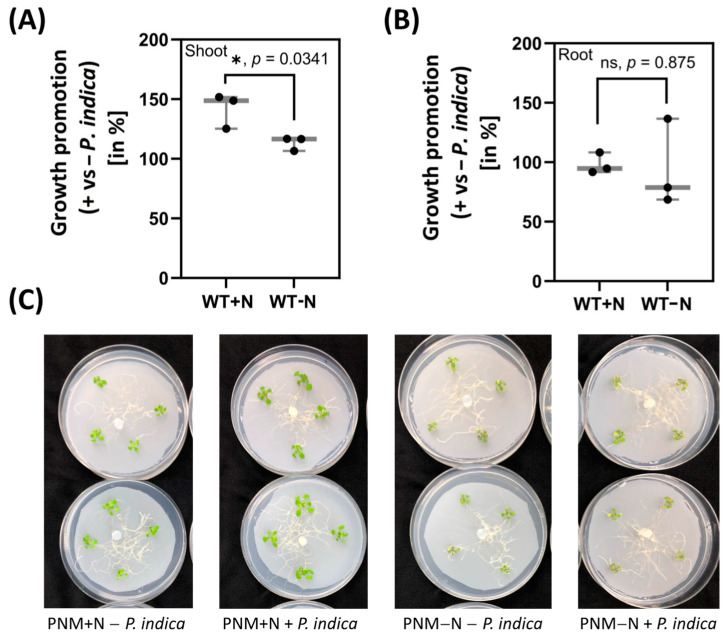
Shoot (**A**) and root (**B**) growth promotion of *A. thaliana* seedlings, which were either grown on full medium (+N) or N-limited medium (−N), in the presence of *P. indica* for 5 days. The % of growth promotion by the fungus was determined for 20 shoots and roots, the plant material grown on the respective media without the fungus was set as 100%. The percentage was determined for each replicate separately since the starting weight of plants was slightly different for the three independent replicates. All three replicates show the same trend of growth promotion. (**C**) shows representative pictures of the co-cultures at harvesting time. Statistic significant differences were analysed using one-way ANOVA (Holm–Sidak test). * significant, ns = not significant.

**Figure 2 ijms-24-15372-f002:**
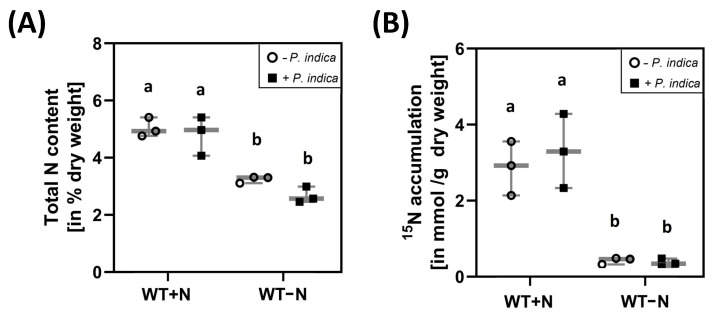
Total N in the shoots of uncolonized (circles) and colonized (squares) *A. thaliana* seedlings. (**A**) Total N in the shoots of seedlings grown with or without *P. indica* on either full or N-limiting conditions for 5 days. The % of N was determined in the dried material of 20 shoots. (**B**) ^15^N accumulation in the shoots of seedlings grown with or without *P. indica* on either full or N-limiting conditions. ^15^N was determined in dried material of 20 shoots. Based on three independent experiments. Statistic significant differences were analysed via one-way ANOVA (Holm–Sidak test). Different small letters indicate statistic significant differences.

**Figure 3 ijms-24-15372-f003:**
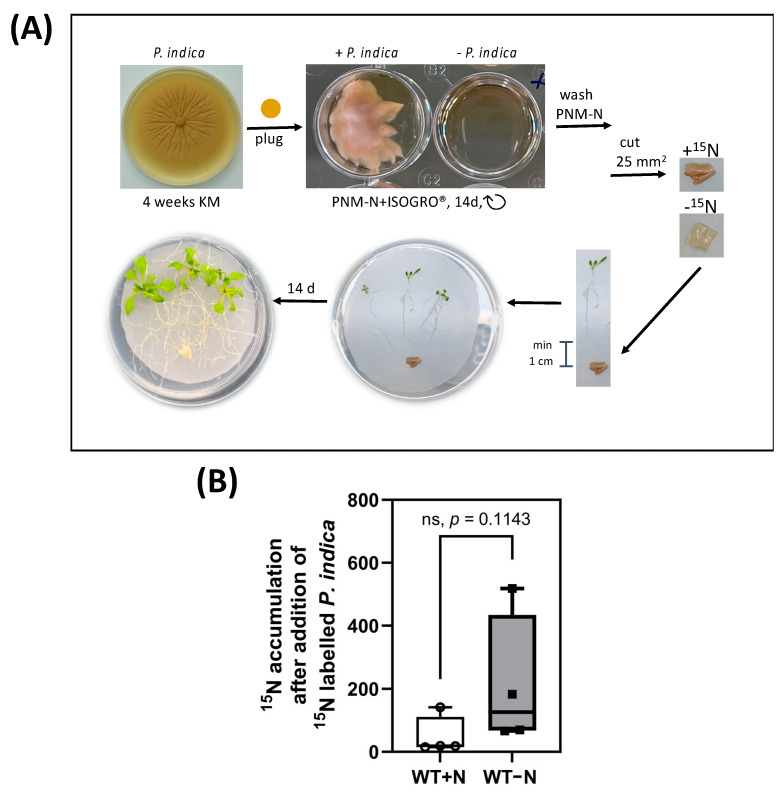
(**A**) Experimental set-up of sterile *A. thaliana* seedlings co-cultivated with ^15^N-labelled *P. indica*. To obtain labelled fungal material, fungal plugs grown on KM plate were transferred to a modified liquid N-free KM medium supplemented with 10 g/L ISOGRO^®^-^15^N and incubated for 14 days in a well plate. The fungal material was separated from the medium, washed carefully with PNM−N and cut into 25 mm^2^ pieces. The fungus was placed onto the nylon membranes with 1 cm distance to the roots. The co-cultures were incubated for 14 days. For further details, see the Section 4. (**B**) ^15^N label in the shoots of seedlings that were exposed to the ^15^N-labelled hyphae on either full (white) or N-limited (grey) media. The accumulation of ^15^N was determined in the dried leaf material of 20 colonized shoots, 14 days after the beginning of the co-culture. For experimental details, see the Section 4. Based on three independent experiments. ns, not significant; analysed via ranked *t*-test (Mann–Whitney test).

**Table 1 ijms-24-15372-t001:** Differentially accumulated metabolites (DAMs) in *A. thaliana* shoots. DAMs regulated by N limitation in *A. thaliana* rosettes without or with *P. indica* colonization. (**A**) amino acids; (**B**) soluble carbohydrates.

(A)	Metabolite RatioLimiting vs. Sufficient N Supply
without *P. indica*	with *P. indica*
Compound	Mean	SE	Mean	SE
Aspartate	0.36	0.07	0.44	0.17
Alanine	0.50	0.06	0.66	0.14
Homoserine	0.69	0.17	0.55	0.10
Glutamine	0.89	0.24	0.64	0.19
Glutamate	0.98	0.21	0.76	0.07
Glycine	1.10	0.46	0.72	0.12
Asparagine	1.22	0.39	0.67	0.11
Proline	1.45	0.19	1.42	0.05
Cystein	1.45	0.30	1.20	0.04
Methionine	1.51	0.71	0.72	0.04
Agmatine(-NH3)	1.64	0.27	1.40	0.52
beta-Alanine	1.67	0.46	1.25	0.12
Threonine	1.86	0.57	2.64	0.73
Valine	2.05	0.59	1.55	0.22
Arginine	2.08	0.86	0.98	0.33
Leucine	2.31	0.50	1.82	0.41
Histidine	2.46	0.88	2.13	0.97
Tyrosine	2.64	0.38	2.46	0.65
Phenylalanine	2.65	0.99	1.46	0.22
Tryptophan	2.70	0.99	2.02	0.48
Lysine	2.78	0.76	2.03	0.65
Isoleucine	3.24	1.10	2.04	0.45
Serine	4.40	0.42	6.00	2.75
**(B)**	**Metabolite Ratio** **Limiting vs. Sufficient N Supply**
**without *P.indica***	**with *P. indica***
**Compound**	**Mean**	**SE**	**Mean**	**SE**
Rhamnose	1.22	0.08	1.20	0.09
Arabinose	1.24	0.19	1.56	0.32
Gentiobiose	1.29	0.17	1.41	0.13
Ribose	1.32	0.07	1.61	0.24
Xylose	1.42	0.32	2.21	0.58
Mannose	1.43	0.28	1.89	0.15
Galactose	1.44	0.34	2.02	0.22
Sucrose	1.55	0.37	2.30	0.77
Glucose	1.59	0.67	3.34	0.73
Fructose	1.72	0.63	3.32	0.75
Maltose	2.22	1.20	1.45	0.29
Trehalose	2.88	2.21	2.02	0.54
Raffinose	3.46	1.55	10.21	3.77
	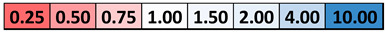

Values are given as the ratio of the (relative) content in N-limiting to N-sufficient growth conditions, as measured by GC-MS profiling. Data are the means +/− SE of three replicates from independent cultures on pools of 20 plantlets. There is no statistical difference with and without *P. indica* with *p* < 0.1. The gradual colour scale is indicated below the table (red, metabolite contents that decreased under limiting N supply; blue, metabolite contents that increased under limiting N supply).

**Table 2 ijms-24-15372-t002:** Differentially regulated transporters in *A. thaliana*. DEGs regulated by nitrogen limitation (−N) in *A. thaliana* tissues without (w/o) or with (w) *P. indica* colonization.

DEGs	Shoot	Root
Category	Gene	Atg Number	−N vs. +N	−N vs. +N	−N vs. +N	−N vs. +N
w/o *P. indica*	w *P. indica*	w/o *P. indica*	w *P. indica*
Nitrate (NRT2 family)	*AtNRT2.2*	At1g08100	x	3.68	x	x
Nitrate (NRT2 family)	*AtNRT2.3*	At5G60780	−1.55	−5.8	x	−5.22
Nitrate (NRT2 family)	*AtNRT2.4*	At5g60770	x	3.61	2.9	2.58
Nitrate (NRT2 family)	*AtNRT2.5*	At1g12940	2.17	3.85	5.63	3.61
Nitrate (NRT2 family)	*AtNRT2.6*	At3g45060	x	x	−3.85	−5.14
Nitrate (NPF family)	*NPF2.6*	At3g45660	x	3.34	x	x
Nitrate (NPF family)	*NPF2.8/NRT1.9*	At5g28470	x	−4.49	x	x
Nitrate (NPF family)	*NPF2.13/NRT1.7*	At1g69870	x	2.48	x	x
Nitrate (NPF family)	*NPF4.1/AIT3*	At3g25260	x	x	x	2.3
Nitrate (NPF family)	*NPF5.3/NRT1.8*	At5g46040	x	4.1	x	x
Nitrate (NPF family)	*NPF5.6*	At2g37900	x	x	−3.57	x
Nitrate (NPF family)	*NPF5.12*	At1g72140	−2.03	x	x	x
Nitrate (NPF family)	*NPF5.14/NRT1.15*	At1g72120	x	1.83	x	x
Nitrate (NPF family)	*NPF6.2/NRT1.4*	At2g26690	x	x	−2.11	−1.61
Ammonium (AMT family)	*AMT1−3*	At3g24300	x	2.42	x	X
Ammonium (AMT family)	*AMT1−4*	At4g28700	x	x	x	2.51
Ammonium (AMT family)	*AMT1−5*	At3g24290	x	4.18	4.12	2.51
Urea	*DUR3*	At5g45380	x	2.46	2.87	2.04
Amino acid (GDU family)	*GDU1*	At4g31730	x	x	−2.1	x
Amino acid (GDU family)	*GDU4*	At2g24762	−1.77	−1.99	−2.54	−1.96
Amino acid (GDU family)	*GDU5*	At5g24920	x	x	−1.8	x
Amino acid (GDU family)	*GDU6*	At3g30725	x	x	−2.89	−2.03
Amino acid (GDU family)	*GDU7*	At5g38770	x	x	−1.81	x
Amino acid (LHT family)	*LHT1*	At5g40780	x	2.16	x	x
Amino acid (LHT family)	*LHT2/AATL2*	At1g24400	x	x	x	−2.06
Amino acid (LHT family)	*LHT3*	At1g61270	x	x	x	1.53
Amino acid (LHT family)	*LHT7*	At4g35180	2.09	2.52	x	X
Amino acid (AAP family)	*AAP3*	At1g77380	x	1.84	x	X
Amino acid (AAP family)	*AAP4*	At5g63850	x	2.15	x	X
Amino acid (AAP family)	*AAP6*	At5g49630	x	1.9	x	X
Amino acid (AAP family)	*AAP7*	At5g23810	x	x	x	1.53
Amino acid (AVT family)	*AVT1E*	At5g02170	x	−4.51	−1.77	X
Amino acid (AVT family)	*AVT1H*	At5g16740	6.41	7.5	2.25	2.72
Amino acid (AVT family)	*AVT3B*	At2g42005	−2.89	−1.65	x	X
Amino acid (CAT family)	*GAT1/BAT1*	At1g08230	x	2.02	x	X
Amino acid (CAT family)	*CAT1/AAT1*	At4g21120	1.65	3.39	x	X
Amino acid (CAT family)	*CAT5*	At2g34960	x	2.18	x	X
Amino acid (UmamiT family)	*UmamiT 4*	At3G18200	x	4.31	x	X
Amino acid (UmamiT family)	*UmamiT 8*	At4G16620	x	1.99	−1.69	X
Amino acid (UmamiT family)	*UmamiT 10*	At3G56620	x	1.89	x	X
Amino acid (UmamiT family)	*UmamiT 13*	At2G37450	x	−2.07	x	−1.74
Amino acid (UmamiT family)	*UmamiT 14*	At2G39510	x	x	−1.54	X
Amino acid (UmamiT family)	*UmamiT 17*	At4G08300	x	1.76	x	X
Amino acid (UmamiT family)	*UmamiT 19*	At1G21890	x	2.27	x	3.02
Amino acid (UmamiT family)	*UmamiT 20*	At4G08290	x	2	−2.01	−1.57
Amino acid (UmamiT family)	*UmamiT 25*	At1G09380	x	2.4	x	X
Amino acid (UmamiT family)	*UmamiT 26*	At1G11460	x	−1.9	x	X
Amino acid (UmamiT family)	*UmamiT 29*	At4G01430	x	1.57	x	X
Amino acid (UmamiT family)	*UmamiT 35*	At1G60050	x	1.75	−2.5	X
Amino acid (UmamiT family)	*UmamiT 36*	At1G70260	x	x	x	1.95
Amino acid (UmamiT family)	*UmamiT 40*	At5G40240	x	2.14	x	X
Amino acid (UmamiT family)	*UmamiT 42*	At5G40210	x	1.84	x	X
Amino acid (UmamiT family)	*UmamiT 43*	At3G28060	x	−2.34	x	X
Amino acid (UmamiT family)	*UmamiT 45*	At3G28100	x	1.85	x	X
Amino acid (UmamiT family)	*UmamiT 46*	At3G28070	x	x	−5.76	X
Amino acid (UmamiT family)	*UmamiT 47*	At3G28080	x	x	−2.94	X
	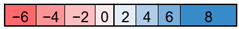

Values are given as log2-fold differential expression identified by RNAseq analysis, *p* < 0.05. x = not differentially expressed compared to full N (+N). The gradual colour scale from red to blue is indicated (red, transcript level decreased under limited N; blue, transcript level increased under limited N supply).

## Data Availability

The dataset is available in the NCBI GEO repository, under accession number GSE239281.

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
