# Peer review of "The Root-Colonizing Endophyte Piriformospora indica Supports Nitrogen-Starved Arabidopsis thaliana Seedlings with Nitrogen Metabolites"

_ijms, 2023, doi:10.3390/ijms242015372_

Round 1
Reviewer 1 Report
The manuscript is well written and addresses an important issue in agriculture. The results are well interpreted.
The authors investigated the potential role of the root-colonizing endophytic fungus Piriformospora indica in contributing to growth under N-limited conditions.The use of microorganisms to promote plant growth and tolerance to abiotic stresses has been a topic of interest to the scientific community. In this manuscript, the authors conducted an indepth study of the involvement of endophytic fungus Piriformospora indica in assisting plant tolerance to N-constrained environments. Therefore, this research contributes to filling the gap in knowledge on plant-microorganism interaction to enhance abiotic stress tolerance. The role of P. indica in promoting N metabolic process has been well document in other crops such as sunflower, barley, Chinese cabbage, rice, chickpea, Arabidopsis, etc. However, research on investigating N metabolism of plants infected with P. indica remains limited. The authors aim to fill this knowledge gap by studying Arabidopsis as a model plant.
The methodology is appropriate. The following are areas that can be further improved.
Page 13. What stage of the plants was RNA extracted?
Page 13. For the DEG analysis, please provide a full list of the samples to be compared (e.g., samples grown on N-limited media VS samples grown on N-supplemented media, etc.)
Page 14. Please provide the reason(s) why discrimination of N from plant or fungus was not possible in root material.
Page 14. Please provide details on the GC-MS analysis. The important detail lacks in the manuscript
I suggest the authors add more literature reviews in the introduction section. The review of literature on this manuscript is too weak to justify the study.
Table 1. Please provide the meaning of the color coding (blue, red, etc.). In addition, the comparison should be supported by some statistical analysis (e.g., paired t-test) which is missing in the methodology.
Table 2. Please provide the meaning of the color coding (blue, red, etc.). Any comparison between root and shoot samples? None of the statistics shown in Figures 1, 2, S1, 3 was described in the methodology. Please provide details on the statistics.
Figure 4 is very confusing. Is this a boxplot? What does the bar for each plot refer to (mean, median, etc,). For each plot, there are only six data points. Please provide a further explanation regarding these six data points. Are these six samples? In addition, there is no statistical analysis for Figure 4.

The quality of English is appropriate.
Author Response
The reply is shown in the attached file.

Reviewer 2 Report
The study conducted by Scholz et al. addresses the regulation of Nitrogen acquisition and plant-growth promotion effect of the P. indica-A. thaliana association. The authors show that the beneficial interaction between these organisms is restricted to N-supply conditions. In addition, they reveal that N metabolites are transported from the fungus to the plant under nitrogen-starving conditions. The evidence is well supported by RNAseq and GC-MS evidence, and the manuscript is clearly written. This manuscript provides valuable information to the field, however, some data must be illustrated with more clarity and transparency, including additional controls and evidence to support their findings.
-The authors should include images of the colonization degree by P. indica in the different roots at the harvesting timepoint. Knowing if the conditions used by the authors alters the colonization kinetics seems crucial to understand the results obtained.
-The authors show that the colonization by the fungus was around 2 times higher under N-limited conditions, although not significantly. This statement is supported by a supplementary figure, where the relative P. indica colonization is shown. However, the figure shows a ratio and not the raw expression values. Showing the raw values seem critical to check the variability obtained of fungus colonization among plants or experiments.
-A reference is missing for this statement in the introduction: “Nitrogen is a key mineral…”
-In Figure 1A and B, the units in the Y axes are referred as relative plant weight (in %). For clarity, the authors must graph the data in g/mg. In addition, instead of bar charts the data should be plotted with box plots, to visualize the distribution of the data. Also, Figure 1 should be accompanied by representative images of the plants.
-Figure 2A. As indicated above, box plots and ug/mg units must be used.
-Figure 3B. “Interestingly, approximately 4-times more…” According to the statistical analysis done by the authors, this increase is not significative. Therefore, the authors can´t conclude that “This indicates that the fungus helps the host…” In this figure, again, for clarity and transparency the authors should use the respective units for such measurement instead of %, and box plots with the datapoints instead of bar charts.
Author Response

(The authors gave the same response as above.)

Reviewer 3 Report
I do find this work interesting and valuable. The data are important in terms of deepening knowledge about possibilities of application such fungi in agriculture for better use of nutrients. The manuscript is quite well written, however there are some issues which should be addressed before publication. Some suggestions for improving are below.
First Page, abstract – second line and in the whole manuscript: please check the proper way of writting „Arabidopsis” and other genera latin names – shouldn’t be they in italics? Besides, you wrote „Arabidopsis” in italics in the title, so you should be consistent.
Page 2 Results, first sentence – I am wondering what this reference [15] is for? What was the reason for that? Does this mean that the data presented in this paragraph were already published? It should be explained (or moved to discussion section).
Figure 3 (A) description – you wrote that you cut fungal material into 5x 5 mm pieces – so, what this 5mm2 on the Figure 3 (A) means? 5x5 mm is 25 mm2
Page 5 – in the end – „Interestingly, approximately 4-times more 15N accumulated in the aerial parts of the plants under N- limiting conditions (Fig. 3B). This indicates that the fungus helps the host with reduced N metabolites to compensate N limitation during growth on NO3--limiting medium” – yes, but differences were not statistically proved!
Page 6 – I did not find Supplementary Table S1 in Suplementary file/s
Table 1 – is this possible to perform statistical analyses for these data (as it was done for previous parameters)?
Figure 4 contains almost the same data as Table 1. Please avoid repetitions.
Page 8 – „Incorporation of 15N into the aerial parts of colonized seedlings is ~4-fold higher under N starvation when compared to seedlings grown on full medium (Fig. 2)” – Figure 2 shows 4-fold lower 15 N accumulation under N starvation – please check?
Conclusion should also contain the main results, not only aspects which need to be investigated in the future. Conclusions and also title should be based on the results of statistical analysis.
Author Response

(The authors gave the same response as above.)
